# Efficient Algorithm for Privately Releasing Smooth Queries

**Ziteng Wang**
Key Laboratory of Machine Perception, MOE
School of EECS
Peking University
wangzt@cis.pku.edu.cn

**Kai Fan**
Key Laboratory of Machine Perception, MOE
School of EECS
Peking University
interfk@hotmail.com

**Jiaqi Zhang**
Key Laboratory of Machine Perception, MOE
School of EECS
Peking University
Zhangjq@cis.pku.edu.cn

**Liwei Wang**
Key Laboratory of Machine Perception, MOE
School of EECS
Peking University
wanglw@cis.pku.edu.cn

## Abstract

We study differentially private mechanisms for answering *smooth* queries on databases consisting of data points in $\mathbb{R}^d$. A $K$-smooth query is specified by a function whose partial derivatives up to order $K$ are all bounded. We develop an $\epsilon$-differentially private mechanism which for the class of $K$-smooth queries has accuracy $O(n^{-\frac{K}{2d+K}}/\epsilon)$. The mechanism first outputs a summary of the database. To obtain an answer of a query, the user runs a public evaluation algorithm which contains no information of the database. Outputting the summary runs in time $O(n^{1+\frac{d}{2d+K}})$, and the evaluation algorithm for answering a query runs in time $\tilde{O}(n^{\frac{d+2+\frac{2d}{K}}{2d+K}})$. Our mechanism is based on $L_\infty$-approximation of (transformed) smooth functions by low degree even trigonometric polynomials with small and efficiently computable coefficients.

## 1 Introduction

Privacy is an important problem in data analysis. Often people want to learn useful information from data that are sensitive. But when releasing statistics of sensitive data, one must tradeoff between the accuracy and the amount of privacy loss of the individuals in the database.

In this paper we consider *differential privacy* [9], which has become a standard concept of privacy. Roughly speaking, a mechanism which releases information about the database is said to preserve differential privacy, if the change of a single database element does not affect the probability distribution of the output significantly. Differential privacy provides strong guarantees against attacks. It ensures that the risk of any individual to submit her information to the database is very small. An adversary can discover almost nothing new from the database that contains the individual's information compared with that from the database without the individual's information. Recently there have been extensive studies of machine learning, statistical estimation, and data mining under the differential privacy framework [29, 5, 18, 17, 6, 30, 20, 4].

Accurately answering statistical queries is an important problem in differential privacy. A simple and efficient method is the Laplace mechanism [9], which adds Laplace noise to the true answers. Laplace mechanism is especially useful for query functions with low sensitivity, which is the maximal difference of the query values of two databases that are different in only one item. A typical

class of queries that has low sensitivity is linear queries, whose sensitivity is $O(1/n)$, where $n$ is the size of the database.

The Laplace mechanism has a limitation. It can answer at most $O(n^2)$ queries. If the number of queries is substantially larger than $n^2$, Laplace mechanism is not able to provide differentially private answers with nontrivial accuracy. Considering that potentially there are many users and each user may submit a set of queries, limiting the number of total queries to be smaller than $n^2$ is too restricted in some situations. A remarkable result due to Blum, Ligett and Roth [2] shows that information theoretically it is possible for a mechanism to answer far more than $n^2$ linear queries while preserving differential privacy and nontrivial accuracy simultaneously.

There are a series of works [10, 11, 21, 16] improving the result of [2]. All these mechanisms are very powerful in the sense that they can answer general and adversely chosen queries. On the other hand, even the fastest algorithms [16, 14] run in time linear in the size of the data universe to answer a query. Often the size of the data universe is much larger than that of the database, so these mechanisms are inefficient. Recently, [25] shows that there is no polynomial time algorithm that can answer $n^{2+o(1)}$ general queries while preserving privacy and accuracy (assuming the existence of one-way function).

Given the hardness result, recently there are growing interests in studying efficient and differentially private mechanisms for restricted class of queries. From a practical point of view, if there exists a class of queries which is rich enough to contain most queries used in applications and allows one to develop fast mechanisms, then the hardness result is not a serious barrier for differential privacy.

One class of queries that attracts a lot of attentions is the $k$-way conjunctions. The data universe for this problem is $\{0,1\}^d$. Thus each individual record has $d$ binary attributes. A $k$-way conjunction query is specified by $k$ features. The query asks what fraction of the individual records in the database has all these $k$ features being 1. A series of works attack this problem using several different techniques [1, 13, 7, 15, 24] . They propose elegant mechanisms which run in time $\text{poly}(n)$ when $k$ is a constant. Another class of queries that yields efficient mechanisms is sparse query. A query is $m$-sparse if it takes non-zero values on at most $m$ elements in the data universe. [3] develops mechanisms which are efficient when $m = \text{poly}(n)$.

When the data universe is $[-1,1]^d$, where $d$ is a constant, [2] considers rectangle queries. A rectangle query is specified by an axis-aligned rectangle. The answer to the query is the fraction of the data points that lie in the rectangle. [2] shows that if $[-1,1]^d$ is discretized to $\text{poly}(n)$ bits of precision, then there are efficient mechanisms for the class of rectangle queries. There are also works studying related range queries [19].

In this paper we study *smooth* queries defined also on data universe $[-1,1]^d$ for constant $d$. A smooth query is specified by a smooth function, which has bounded partial derivatives up to a certain order. The answer to the query is the average of the function values on data points in the database. Smooth functions are widely used in machine learning and data analysis [28]. There are extensive studies on the relation between smoothness, regularization, reproducing kernels and generalization ability [27, 22].

Our main result is an $\epsilon$-differentially private mechanism for the class of $K$-smooth queries, which are specified by functions with bounded partial derivatives up to order $K$. The mechanism has $(\alpha, \beta)$-accuracy, where $\alpha = O(n^{-\frac{K}{2d+K}}/\epsilon)$ for $\beta \geq e^{-O(n^{\frac{d}{2d+K}})}$. The mechanism first outputs a summary of the database. To obtain an answer of a smooth query, the user runs a public evaluation procedure which contains no information of the database. Outputting the summary has running time $O\left(n^{1+\frac{d}{2d+K}}\right)$, and the evaluation procedure for answering a query runs in time $\tilde{O}(n^{\frac{d+2+\frac{2d}{K}}{2d+K}})$. The mechanism has the advantage that both the accuracy and the running time for answering a query improve quickly as $K/d$ increases (see also Table 1 in Section 3).

Our algorithm is a $L_\infty$-approximation based mechanism and is motivated by [24], which considers approximation of $k$-way conjunctions by low degree polynomials. The basic idea is to approximate the whole query class by linear combination of a small set of basis functions. The technical difficulties lie in that in order that the approximation induces an efficient and differentially private mechanism, all the linear coefficients of the basis functions must be small and efficiently computable. To guarantee these properties, we first transform the query function. Then by using even trigono-

metric polynomials as basis functions we prove a constant upper bound for the linear coefficients. The smoothness of the functions also allows us to use an efficient numerical method to compute the coefficients to a precision so that the accuracy of the mechanism is not affected significantly.

## 2  Background

Let $D$ be a database containing $n$ data points in the data universe $\mathcal{X}$. In this paper, we consider the case that $\mathcal{X} \subset \mathbb{R}^d$ where $d$ is a constant. Typically, we assume that the data universe $\mathcal{X} = [-1, 1]^d$. Two databases $D$ and $D'$ are called neighbors if $|D| = |D'| = n$ and they differ in exactly one data point. The following is the formal definition of differential privacy.

**Definition 2.1** (($\epsilon, \delta$)-differential privacy)**.** A sanitizer $\mathcal{S}$ which is an algorithm that maps input database into some range $\mathcal{R}$ is said to preserve ($\epsilon, \delta$)-differential privacy, if for all pairs of neighbor databases $D, D'$ and for any subset $A \subset \mathcal{R}$, it holds that

$$\mathbb{P}(\mathcal{S}(D) \in A) \leq \mathbb{P}(\mathcal{S}(D') \in A) \cdot e^\epsilon + \delta.$$

If $\mathcal{S}$ preserves ($\epsilon, 0$)-differential privacy, we say $\mathcal{S}$ is $\epsilon$-differentially private.

We consider *linear queries*. Each linear query $q_f$ is specified by a function $f$ which maps data universe $[-1, 1]^d$ to $\mathbb{R}$, and $q_f$ is defined by $q_f(D) := \frac{1}{|D|} \sum_{x \in D} f(x)$.

Let $Q$ be a set of queries. The accuracy of a mechanism with respect to $Q$ is defined as follows.

**Definition 2.2** (($\alpha, \beta$)-accuracy)**.** Let $Q$ be a set of queries. A sanitizer $\mathcal{S}$ is said to have ($\alpha, \beta$)-accuracy for size $n$ databases with respect to $Q$, if for every database $D$ with $|D| = n$ the following holds

$$\mathbb{P}(\exists q \in Q, \quad |\mathcal{S}(D, q) - q(D)| \geq \alpha) \leq \beta,$$

where $\mathcal{S}(D, q)$ is the answer to $q$ given by $\mathcal{S}$.

We will make use of Laplace mechanism [9] in our algorithm. Laplace mechanism adds Laplace noise to the output. We denote by $\mathrm{Lap}(\sigma)$ the random variable distributed according to the Laplace distribution with parameter $\sigma$: $\mathbb{P}(\mathrm{Lap}(\sigma) = x) = \frac{1}{2\sigma} \exp(-|x|/\sigma)$.

We will design a differentially private mechanism which is accurate with respect to a query set $Q$ possibly consisting of infinite number of queries. Given a database $D$, the sanitizer outputs a *summary* which preserves differential privacy. For any $q_f \in Q$, the user makes use of an evaluation procedure to measure $f$ on the summary and obtain an approximate answer of $q_f(D)$. Although we may think of the evaluation procedure as part of the mechanism, it does not contain any information of the database and therefore is public. We will study the running time for the sanitizer outputting the summary. Ideally it is $O(n^c)$ for some constant $c$ not much larger than 1. For the evaluation procedure, the running time *per query* is the focus. Ideally it is sublinear in $n$. Here and in the rest of the paper, we assume that calculating the value of $f$ on a data point $x$ can be done in unit time.

In this work we will frequently use *trigonometric polynomials*. For the univariate case, a function $p(\theta)$ is called a trigonometric polynomial of degree $m$ if $p(\theta) = a_0 + \sum_{l=1}^{m} (a_l \cos l\theta + b_l \sin l\theta)$, where $a_l, b_l$ are constants. If $p(\theta)$ is an even function, we say that it is an even trigonometric polynomial, and $p(\theta) = a_0 + \sum_{l=1}^{m} a_l \cos l\theta$. For the multivariate case, if $p(\theta_1, \ldots, \theta_d) = \sum_{\mathbf{l}=(l_1,\ldots,l_d)} a_{\mathbf{l}} \cos(l_1\theta_1) \ldots \cos(l_d\theta_d)$, then $p$ is said to be an even trigonometric polynomial (with respect to each variable), and the degree of $\theta_i$ is the upper limit of $l_i$.

## 3  Efficient differentially private mechanism

Let us first describe the set of queries considered in this work. Since each query $q_f$ is specified by a function $f$, a set of queries $Q_F$ can be specified by a set of functions $F$. Remember that each $f \in F$ maps $[-1, 1]^d$ to $\mathbb{R}$. For any point $\mathbf{x} = (x_1, \ldots, x_d) \in [-1, 1]^d$, if $\mathbf{k} = (k_1, \ldots, k_d)$ is a $d$-tuple with nonnegative integers, then we define

$$D^{\mathbf{k}} := D_1^{k_1} \cdots D_d^{k_d} := \frac{\partial^{k_1}}{\partial x_1^{k_1}} \cdots \frac{\partial^{k_d}}{\partial x_d^{k_d}}.$$

```
Parameters: Privacy parameters $\epsilon, \delta > 0$; Failure probability $\beta > 0$;
          Smoothness order $K \in \mathbb{N}$; Set $t = n^{\frac{1}{2d+K}}$.
Input: Database $D \in \left( [-1, 1]^d \right)^n$.
Output: A $t^d$-dimensional vector as the summary.
Algorithm:
    For each $\mathbf{x} = (x_1, \dots, x_d) \in D$:
        Set:  $\theta_i(\mathbf{x}) = \arccos(x_i)$, $i = 1, \dots, d$;
    For every $d$-tuple of nonnegative integers $\mathbf{m} = (m_1, \dots, m_d)$, where $\|\mathbf{m}\|_\infty \le t - 1$
        Compute: $\mathrm{Su}_\mathbf{m}(D) = \frac{1}{n} \sum_{\mathbf{x} \in D} \cos(m_1 \theta_1(\mathbf{x})) \dots \cos(m_d \theta_d(\mathbf{x}))$;
        $\widehat{\mathrm{Su}}_\mathbf{m}(D) \leftarrow \mathrm{Su}_\mathbf{m}(D) + \mathrm{Lap}\left( \frac{t^d}{n\epsilon} \right)$ ;
    Let $\widehat{\mathrm{Su}}(D) = \left( \widehat{\mathrm{Su}}_\mathbf{m}(D) \right)_{\|\mathbf{m}\|_\infty \le t-1}$ be a $t^d$ dimensional vector;
    Return: $\widehat{\mathrm{Su}}(D)$.
```

**Algorithm 1:** Outputting the summary

```
Parameters: $t = n^{\frac{1}{2d+K}}$.
Input: A query $q_f$, where $f : [-1, 1]^d \to \mathbb{R}$ and $f \in C_B^K$,
        Summary $\widehat{\mathrm{Su}}(D)$ (a $t^d$-dimensional vector).
Output: Approximate answer to $q_f(D)$.
Algorithm:
    Let $g_f(\boldsymbol{\theta}) = f(\cos(\theta_1), \dots, \cos(\theta_d))$, $\boldsymbol{\theta} = (\theta_1, \dots, \theta_d) \in [-\pi, \pi]^d$;
    Compute a trigonometric polynomial approximation $p_t(\boldsymbol{\theta})$ of $g_f(\boldsymbol{\theta})$,
        where the degree of each $\theta_i$ is $t$;    // see Section 4 for details of computation.
    Denote $p_t(\boldsymbol{\theta}) = \sum_{\mathbf{m}=(m_1,\dots,m_d), \|\mathbf{m}\|_\infty < t} c_\mathbf{m} \cos(m_1 \theta_1) \dots \cos(m_d \theta_d)$;
    Let $\mathbf{c} = (c_\mathbf{m})_{\|\mathbf{m}\|_\infty < t}$ be a $t^d$-dimensional vector;
    Return: the inner product $< \mathbf{c}, \widehat{\mathrm{Su}}(D) >$.
```

**Algorithm 2:** Answering a query

Let $|\mathbf{k}| := k_1 + \dots + k_d$. Define the $K$-norm as

$$\|f\|_K := \sup_{|\mathbf{k}| \le K} \sup_{\mathbf{x} \in [-1,1]^d} |D^\mathbf{k} f(\mathbf{x})|.$$

We will study the set $C_B^K$ which contains all *smooth* functions whose derivatives up to order $K$ have $\infty$-norm upper bounded by a constant $B > 0$. Formally, $C_B^K := \{ f : \|f\|_K \le B \}$. The set of queries specified by $C_B^K$, denoted as $Q_{C_B^K}$, is our focus. Smooth functions have been studied in depth in machine learning [26, 28, 27] and found wide applications [22].

The following theorem is our main result. It says that if the query class is specified by smooth functions, then there is a very efficient mechanism which preserves $\epsilon$-differential privacy and good accuracy. The mechanism consists of two parts: One for outputting a summary of the database, the other for answering a query. The two parts are described in Algorithm 1 and Algorithm 2 respectively. The second part of the mechanism contains no private information of the database.

**Theorem 3.1.** *Let the query set be $Q_{C_B^K} = \{ q_f = \frac{1}{n} \sum_{\mathbf{x} \in D} f(\mathbf{x}) : f \in C_B^K \}$, where $K \in \mathbb{N}$ and $B > 0$ are constants. Let the data universe be $[-1, 1]^d$, where $d \in \mathbb{N}$ is a constant. Then the mechanism $\mathcal{S}$ given in Algorithm 1 and Algorithm 2 satisfies that for any $\epsilon > 0$, the following hold:*

*1) The mechanism is $\epsilon$-differentially private.*

*2) For any $\beta \ge 10 \cdot e^{-\frac{1}{5}(n^{\frac{d}{2d+K}})}$ the mechanism is $(\alpha, \beta)$-accurate, where $\alpha = O\left( \left(\frac{1}{n}\right)^{\frac{K}{2d+K}} / \epsilon \right)$, and the hidden constant depends only on $d$, $K$ and $B$.*

Table 1: Performances vs. Order of smoothness

| Order of smoothness | Accuracy $\alpha$ | Time: Outputting summary | Time: Answering a query |
|---|---|---|---|
| $K = 1$ | $O((\frac{1}{n})^{\frac{1}{2d+1}})$ | $O(n^{\frac{3}{2}})$ | $\tilde{O}(n^{\frac{3}{2}+\frac{1}{4d+2}})$ |
| $K = 2d$ | $O(\frac{1}{\sqrt{n}})$ | $O(n^{\frac{5}{4}})$ | $\tilde{O}(n^{\frac{1}{4}+\frac{3/4}{d}})$ |
| $\frac{d}{K} = \epsilon_0 \ll 1$ | $O((\frac{1}{n})^{1-2\epsilon_0})$ | $O(n^{1+\epsilon_0})$ | $\tilde{O}(n^{\epsilon_0(1+\frac{3}{d})})$ |

3) The running time for $\mathcal{S}$ to output the summary is $O(n^{\frac{3d+K}{2d+K}})$.

4) The running time for $\mathcal{S}$ to answer a query is $O(n^{\frac{d+2+\frac{2d}{K}}{2d+K}} \text{polylog}(n))$.

The proof of Theorem 3.1 is given in the supplementary material. To have a better idea of how the performances depend on the order of smoothness, let us consider three cases. The first case is $K = 1$, i.e., the query functions only have the first order derivatives. Another extreme case is $K \gg d$, and we assume $d/K = \epsilon_0 \ll 1$. We also consider a case in the middle by assuming $K = 2d$. Table 1 gives simplified upper bounds for the error and running time in these cases. We have the following observations:

1) The accuracy $\alpha$ improves dramatically from roughly $O(n^{-\frac{1}{2d}})$ to nearly $O(n^{-1})$ as $K$ increases. For $K > 2d$, the error is smaller than the sampling error $O(\frac{1}{\sqrt{n}})$.

2) The running time for outputting the summary does not change too much, because reading through the database requires $\Omega(n)$ time.

3) The running time for answering a query reduces significantly from roughly $O(n^{3/2})$ to nearly $O(n^{\epsilon_0})$ as $K$ getting large. When $K = 2d$, it is about $n^{1/4}$ if $d$ is not too small. In practice, the speed for answering a query may be more important than that for outputting the summary since the sanitizer only output the summary once. Thus having an $n^c$-time ($c \ll 1$) algorithm for query answering will be appealing.

Conceptually our mechanism is simple. First, by change of variables we have $g_f(\theta_1, \ldots, \theta_d) = f(\cos\theta_1, \ldots, \cos\theta_d)$. It also transforms the data universe from $[-1, 1]^d$ to $[-\pi, \pi]^d$. Note that for each variable $\theta_i$, $g_f$ is an even function. To compute the summary, the mechanism just gives noisy answers to queries specified by *even trigonometric monomials* $\cos(m_1\theta_1)\ldots\cos(m_d\theta_d)$. For each trigonometric monomial, the highest degree of any variable is $t := \max_d m_d = O(n^{\frac{1}{2d+K}})$. The summary is a $O(n^{\frac{d}{2d+K}})$-dimensional vector. To answer a query specified by a smooth function $f$, the mechanism computes a trigonometric polynomial approximation of $g_f$. The answer to the query $q_f$ is a linear combination of the summary by the coefficients of the approximation trigonometric polynomial.

Our algorithm is an $L_\infty$-approximation based mechanism, which is motivated by [24]. An approximation based mechanism relies on three conditions: 1) There exists a small set of basis functions such that every query function can be well approximated by a linear combination of them; 2) All the linear coefficients are small; 3) The whole set of the linear coefficients can be computed efficiently.

If these conditions hold, then the mechanism just outputs noisy answers to the set of queries specified by the basis functions as the summary. When answering a query, the mechanism computes the coefficients with which the linear combination of the basis functions approximate the query function. The answer to the query is simply the inner product of the coefficients and the summary vector.

The following theorem guarantees that by change of variables and using even trigonometric polynomials as the basis functions, the class of smooth functions has all the three properties described above.

**Theorem 3.2.** *Let $\gamma > 0$. For every $f \in C_B^K$ defined on $[-1, 1]^d$, let*

$$g_f(\theta_1, \ldots, \theta_d) = f(\cos\theta_1, \ldots, \cos\theta_d), \qquad \theta_i \in [-\pi, \pi].$$

*Then, there is an even trigonometric polynomial p whose degree of each variable is $t(\gamma) = \left(\frac{1}{\gamma}\right)^{1/K}$:*

$$p(\theta_1, \ldots, \theta_d) = \sum_{0 \leq l_1, \ldots, l_d < t(\gamma)} c_{l_1, \ldots, l_d} \cos(l_1 \theta_1) \ldots \cos(l_d \theta_d),$$

*such that*

*1) $\|g_f - p\|_\infty \leq \gamma$.*

*2) All the linear coefficients $c_{l_1, \ldots, l_d}$ can be uniformly upper bounded by a constant $M$ independent of $t(\gamma)$ (i.e., $M$ depends only on $K$, $d$, and $B$).*

*3) The whole set of the linear coefficients can be computed in time $O\left(\left(\frac{1}{\gamma}\right)^{\frac{d+2}{K} + \frac{2d}{K^2}} \cdot \mathrm{polylog}(\frac{1}{\gamma})\right)$.*

Theorem 3.2 is proved in Section 4. Based on Theorem 3.2, the proof of Theorem 3.1 is mainly the argument for Laplace mechanism together with an optimization of the approximation error $\gamma$ trading-off with the Laplace noise. (Please see the supplementary material.)

# 4 $L_\infty$-approximation of smooth functions: small and efficiently computable coefficients

In this section we prove Theorem 3.2. That is, for every $f \in C_B^K$ the corresponding $g_f$ can be approximated by a low degree trigonometric polynomial in $L_\infty$-norm. We also require that the linear coefficients of the trigonometric polynomial are all small and can be computed efficiently. These properties are crucial for the differentially private mechanism to be accurate and efficient.

In fact, $L_\infty$-approximation of smooth functions in $C_B^K$ by polynomial (and other basis functions) is an important topic in approximation theory. It is well-known that for every $f \in C_B^K$ there is a low degree polynomial with small approximation error. However, it is not clear whether there is an upper bound for the linear coefficients that is sufficiently good for our purpose. Instead we transform $f$ to $g_f$ and use trigonometric polynomials as the basis functions in the mechanism. Then we are able to give a constant upper bound for the linear coefficients. We also need to compute the coefficients efficiently. But results from approximation theory give the coefficients as complicated integrals. We adopt an algorithm which fully exploits the smoothness of the function and thus can efficiently compute approximations of the coefficients to certain precision so that the errors involved do not affect the accuracy of the differentially private mechanism too much.

Below, Section 4.1 describes the classical theory on trigonometric polynomial approximation of smooth functions. Section 4.2 shows that the coefficients have a small upper bound and can be efficiently computed. Theorem 3.2 then follows from these results.

## 4.1 Trigonometric polynomial approximation with generalized Jackson kernel

This section mainly contains known results of trigonometric polynomial approximation, stated in a way tailored to our problem. For a comprehensive description of univariate approximation theory, please refer to the excellent book of [8]; and to [23] for multivariate approximation theory.

Let $g_f$ be the function obtained from $f \in C_B^K([-1, 1]^d)$: $g_f(\theta_1, \ldots, \theta_d) = f(\cos\theta_1, \ldots, \cos\theta_d)$. Note that $g_f \in C_{B'}^K([-\pi, \pi]^d)$ for some constant $B'$ depending only on $B, K, d$, and $g_f$ is even with respect to each variable. The key tool in trigonometric polynomial approximation of smooth functions is the generalized Jackson kernel.

**Definition 4.1.** Define the generalized Jackson kernel as $J_{t,r}(s) = \frac{1}{\lambda_{t,r}} \left(\frac{\sin(ts/2)}{\sin(s/2)}\right)^{2r}$, where $\lambda_{t,r}$ is determined by $\int_{-\pi}^{\pi} J_{t,r}(s)\mathrm{d}s = 1$.

$J_{t,r}(s)$ is an even trigonometric polynomial of degree $r(t-1)$. Let $H_{t,r}(s) = J_{t',r}(s)$, where $t' = \lfloor t/r \rfloor + 1$. Then $H_{t,r}$ is an even trigonometric polynomial of degree at most $t$. We write

$$H_{t,r}(s) = a_0 + \sum_{l=1}^{t} a_l \cos ls. \tag{1}$$

Suppose that $g$ is a univariate function defined on $[-\pi, \pi]$ which satisfies that $g(-\pi) = g(\pi)$. Define the approximation operator $I_{t,K}$ as

$$I_{t,K}(g)(x) = -\int_{-\pi}^{\pi} H_{t,r}(s) \sum_{l=1}^{K+1} (-1)^l \binom{K+1}{l} g(x+ls)\mathrm{d}s, \tag{2}$$

where $r = \lceil \frac{K+3}{2} \rceil$. It is not difficult to see that $I_{t,K}$ maps $g$ to a trigonometric polynomial of degree at most $t$.

Next suppose that $g$ is a $d$-variate function defined on $[-\pi, \pi]^d$, and is even with respect to each variable. Define an operator $I_{t,K}^d$ as sequential composition of $I_{t,K,1}, \ldots, I_{t,K,d}$, where $I_{t,K,j}$ is the approximation operator given in (2) with respect to the $j$th variable of $g$. Thus $I_{t,K}^d(g)$ is a trigonometric polynomial of $d$-variables and each variable has degree at most $t$.

**Theorem 4.1.** *Suppose that $g$ is a $d$-variate function defined on $[-\pi, \pi]^d$, and is even with respect to each variable. Let $D_j^{(K)}g$ be the $K$th order partial derivative of $g$ respect to the $j$-th variable. If $\|D_j^{(K)}g\|_\infty \leq M$ for some constant $M$ for all $1 \leq j \leq d$, then there is a constant $C$ such that*

$$\|g - I_{t,K}^d(g)\|_\infty \leq \frac{C}{t^{K+1}},$$

*where $C$ depends only on $M$, $d$ and $K$.*

## 4.2 The linear coefficients

In this subsection we study the linear coefficients in the trigonometric polynomial $I_{t,K}^d(g_f)$. The previous subsection established that $g_f$ can be approximated by $I_{t,K}^d(g_f)$ for a small $t$. Here we consider the upper bound and approximate computation of the coefficients. Since $I_{t,K}^d(g_f)(\theta_1, \ldots, \theta_d)$ is even with respect to each variable, we write

$$I_{t,K}^d(g_f)(\theta_1, \ldots, \theta_d) = \sum_{0 \leq n_1, \ldots, n_d \leq t} c_{n_1, \ldots, n_d} \cos(n_1\theta_1) \ldots \cos(n_d\theta_d). \tag{3}$$

**Fact 4.2.** *The coefficients $c_{n_1, \ldots, n_d}$ of $I_{t,K}^d(g_f)$ can be written as*

$$c_{n_1, \ldots, n_d} = (-1)^d \sum_{\substack{1 \leq k_1, \ldots, k_d \leq K+1 \\ 0 \leq l_1, \ldots, l_d \leq t \\ l_i = k_i \cdot n_i \forall i \in [d]}} m_{l_1, k_1, \ldots, l_d, k_d}, \tag{4}$$

*where*

$$m_{l_1, k_1, \ldots, l_d, k_d} = \prod_{i=1}^{d} (-1)^{k_i} a_{l_i} \binom{K+1}{k_i} \left( \int_{[-\pi, \pi]^d} \prod_{i=1}^{d} \cos\left(\frac{l_i}{k_i}\theta_i\right) g_f(\boldsymbol{\theta})\mathrm{d}\boldsymbol{\theta} \right), \tag{5}$$

*and $a_{l_i}$ is the linear coefficient of $\cos(l_i s)$ in $H_{t,r}(s)$ as given in (1).*

The following lemma shows that the coefficients $c_{n_1, \ldots, n_d}$ of $I_{t,K}^d(g_f)$ can be uniformly upper bounded by a constant independent of $t$.

**Lemma 4.3.** *There exists a constant $M$ which depends only on $K, B, d$ but independent of $t$, such that for every $f \in C_B^K$, all the linear coefficients $c_{n_1, \ldots, n_d}$ of $I_{t,K}^d(g_f)$ satisfy*

$$|c_{n_1, \ldots, n_d}| \leq M.$$

The proof of Lemma 4.3 is given in the supplementary material. Now we consider the computation of the coefficients $c_{n_1, \ldots, n_d}$ of $I_{t,K}^d(g_f)$. Note that each coefficient involves $d$-dimensional integrations of smooth functions, so we have to numerically compute approximations of them. For function class $C_B^K$ defined on $[-1, 1]^d$, traditional numerical integration methods run in time $O((\frac{1}{\tau})^{d/K})$ in order that the error is less than $\tau$. Here we adopt the sparse grids algorithm due to Gerstner and Griebel [12] which fully exploits the smoothness of the integrand. By choosing a particular quadrature rule as the algorithm's subroutine, we are able to prove that the running time of the sparse grids

is bounded by $O((\frac{1}{\tau})^{2/K})$. The sparse grids algorithm, the theorem giving the bound for the running time and its proof are all given in the supplementary material. Based on these results, we establish the running time for computing the approximate coefficients of the trigonometric polynomial, which is stated in the following Lemma.

**Lemma 4.4.** *Let $\hat{c}_{n_1,\ldots,n_d}$ be an approximation of the coefficient $c_{n_1,\ldots,n_d}$ of $I_{t,K}^d(g_f)$ obtained by approximately computing the integral in* (5) *with a version of the sparse grids algorithm [12] (given in the supplementary material). Let*

$$\hat{I}_{t,K}^d(g_f)(\theta_1,\ldots,\theta_d) = \sum_{0 \leq n_1,\ldots,n_d \leq t} \hat{c}_{n_1,\ldots,n_d} \cos(n_1\theta_1)\ldots\cos(n_d\theta_d).$$

*Then for every $f \in C_B^K$, in order that $\|\hat{I}_{t,K}^d(g_f) - I_{t,K}^d(g_f)\|_\infty \leq O\left(t^{-K}\right)$, it suffices that the computation of all the coefficients $\hat{c}_{n_1,\ldots,n_d}$ runs in time $O\left(t^{(1+\frac{2}{K})d+2} \cdot \mathrm{polylog}(t)\right)$. In addition, $\max_{n_1,\ldots,n_d} |\hat{c}_{n_1,\ldots,n_d} - c_{n_1,\ldots,n_d}| = o(1)$ as $t \to \infty$.*

The proof of Lemma 4.4 is given in the supplementary material. Theorem 3.2 then follows easily from Lemma 4.3 and Lemma 4.4.

***Proof of Theorem 3.2.*** Setting $t = t(\gamma) = \left(\frac{1}{\gamma}\right)^{1/K}$. Let $p = \hat{I}_{m,K}^d(g_f)$. Combining Lemma 4.3 and Lemma 4.4, and note that the coefficients $\hat{c}_{n_1,\ldots,n_d}$ are upper bounded by a constant, the theorem follows.

$\square$

## 5    Conclusion

In this paper we propose an $\epsilon$-differentially private mechanism for efficiently releasing $K$-smooth queries. The accuracy of the mechanism is $O((\frac{1}{n})^{\frac{K}{2d+K}})$. The running time for outputting the summary is $O(n^{1+\frac{d}{2d+K}})$, and is $\tilde{O}(n^{\frac{d+2+2d/K}{2d+K}})$ for answering a query. The result can be generalized to $(\epsilon,\delta)$-differential privacy straightforwardly using the composition theorem [11]. The accuracy improves slightly to $O((\frac{1}{n})^{\frac{2K}{3d+2K}} \log(\frac{1}{\delta})^{\frac{K}{3d+2K}})$, while the running time for outputting the summary and answering the query increase slightly. Our mechanism is based on approximation of smooth functions by linear combination of a small set of basis functions with small and efficiently computable coefficients. Directly approximating functions in $C_B^K([-1,1]^d)$ by polynomials does not guarantee small coefficients and is less efficient. To achieve these goals we use trigonometric polynomials to approximate a transformation of the query functions.

It is worth pointing out that the approximation considered here for differential privacy is $L_\infty$-approximation, because the accuracy is defined in the worst case sense with respect to databases and queries. $L_\infty$-approximation is different to $L_2$-approximation, which is simply the Fourier transform if we use trigonometric polynomials as the basis functions. $L_2$-approximation does not guarantee (worst case) accuracy.

For the class of smooth functions defined on $[-1,1]^d$ where $d$ is a constant, in fact it is not difficult to design a $\mathrm{poly}(n)$ time differentially private mechanism. One can discretize $[-1,1]^d$ to $O(\frac{1}{\sqrt{n}})$ precision, and use the differentially private mechanism for answering general queries (e.g., [16]). However the mechanism runs in time $\tilde{O}(n^{d/2})$ to answer a query, and provides $\tilde{O}(n^{-1/2})$ accuracy. In contrast our mechanism exploits higher order smoothness of the queries. It is always more efficient, and for queries highly smooth it is more accurate.

### Acknowledgments

This work was supported by NSFC(61222307, 61075003) and a grant from MOE-Microsoft Key Laboratory of Statistics and Information Technology of Peking University. We also thank Di He for very helpful discussions.

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
