[Supplementary Material]

# Supplementary Material — Proof of the Theorems

**Ziteng Wang**
Key Laboratory of Machine Perception, MOE
School of EECS
Peking University
wangzt@cis.pku.edu.cn

**Kai Fan**
Key Laboratory of Machine Perception, MOE
School of EECS
Peking University
interfk@hotmail.com

**Jiaqi Zhang**
Key Laboratory of Machine Perception, MOE
School of EECS
Peking University
Zhangjq@cis.pku.edu.cn

**Liwei Wang**
Key Laboratory of Machine Perception, MOE
School of EECS
Peking University
wanglw@cis.pku.edu.cn

Here we give proofs of the theorems and lemmas, as well as a description of the sparse grids algorithm.

## A   Proof of Theorem 3.1

***Proof of Theorem 3.1.***   We prove the four results separately.

1) The summary is a $t^d$-dimensional vector with sensitivity $\frac{t^d}{n}$. By the standard argument for Laplace mechanism, adding $t^d$ i.i.d. Laplace noise $\mathrm{Lap}(\frac{t^d}{n\epsilon})$ preserves $\epsilon$-differential privacy.

2) The error of the answer to each query consists of two parts: the approximation error and the noise error. Setting the approximation error $\gamma$ in Theorem 3.2 as $\gamma = n^{-\frac{K}{2d+K}}$. Then the degree of each variable in $g(\boldsymbol{\theta})$ is $t(\gamma) = \left(\frac{1}{\gamma}\right)^{1/K} = n^{\frac{1}{2d+K}}$, which is the same as $t$ given in Algorithm 1. Now consider the error induced by the Laplace noise. The noise error is simply the inner product of the $t^d$ linear coefficients $c_{l_1,\dots,l_d}$ and $t^d$ i.i.d. $\mathrm{Lap}(\frac{t^d}{n\epsilon})$. Since the coefficients are uniformly bounded by a constant, the noise error is bounded by the sum of $t^d$ independent and exponentially distributed random variables (i.e., $|\mathrm{Lap}(\frac{t^d}{n\epsilon})|$). The following lemma gives it an upper bound.

**Lemma A.1.** *Let $X_1, \dots, X_N$ be i.i.d. random variables with p.d.f. $\mathbb{P}(X_i = x) = \frac{1}{\sigma}e^{-x/\sigma}$ for $x \geq 0$. Then*

$$\mathbb{P}(\sum_{i=1}^{N} X_i \geq 2N\sigma) \leq 10 \cdot e^{-\frac{N}{5}}.$$

*Proof.*   Let $Y = \sum_{i=1}^{N} X_i$. It is well-known that $Y$ satisfies the gamma distribution, and for $\forall u > 0$

$$\mathbb{P}(Y \geq u) \leq e^{-\frac{u}{\sigma}} \sum_{n=0}^{N-1} \frac{1}{n!} \left(\frac{u}{\sigma}\right)^n.$$

Thus

$$\mathbb{P}(Y \geq 2N\sigma) \leq e^{-2N} \sum_{n=0}^{N-1} \frac{1}{n!}(2N)^n.$$

Note that for $n < N$

$$\frac{\frac{1}{n!}(2N)^n}{e^{2N}} \leq \frac{\frac{1}{N!}(2N)^N}{\frac{1}{(2N)!}(2N)^{2N}} \leq \prod_{n=1}^{N-1}\left(1 - \frac{n}{2N}\right) \leq e^{-\frac{N-1}{4}}.$$

Thus

$$\mathbb{P}(Y \geq 2N\sigma) \leq e^{-2N}\left(e^{2N}Ne^{-\frac{N-1}{4}}\right) \leq 10 \cdot e^{-\frac{N}{5}}.$$

$\square$

Part 2) of Theorem 3.1 then follows from Lemma A.1.

3) This is straightforward since the summary is a $t^d$-dimensional vector and for each item the running time is $O(n)$.

4) According to our setting of $t$, it is easy to check that the error induced by Laplace noise and that of approximation have the same order. Then by the third part of Theorem 3.2 we have the running time for computing the coefficients of the trigonometric polynomial is $O\left(n^{\frac{d+2+\frac{2d}{K}}{2d+K}} \cdot \text{polylog}(n)\right)$.

The result follows since computing the inner product has running time $O(n^{\frac{d}{2d+K}})$, which is much less than computing the coefficients.

$\square$

## B  Proof of Lemma 4.3

We first give a simple lemma.

**Lemma B.1.** *Let*

$$H_{t,r}(s) = \sum_{l=0}^{t} a_l \cos ls. \tag{1}$$

*Then for all $l = 0, 1, \ldots, t$*

$$|a_l| \leq 1/\pi.$$

*Proof.* For any $l \in \{0, 1, \ldots, t\}$, multiplying $\cos ls$ on both sides of (1) and integrating from $-\pi$ to $\pi$, we obtain that for some $\xi \in [-\pi, \pi]$,

$$a_l = \frac{1}{\pi}\int_{-\pi}^{\pi} H_{t,r}(s)\cos ls\,\mathrm{d}s = \frac{\cos l\xi}{\pi}\int_{-\pi}^{\pi} H_{t,r}(s)\,\mathrm{d}s = \frac{\cos l\xi}{\pi}.$$

where in the last equation we use the identity

$$\int_{-\pi}^{\pi} H_{t,r}(s)\,\mathrm{d}s = 1.$$

This completes the proof.

$\square$

***Proof of Lemma 4.3.*** We first bound $m_{l_1,k_1,\ldots,l_d,k_d}$. Recall that (see also (5) in Fact 4.2)

$$m_{l_1,k_1,\ldots,l_d,k_d} = \prod_{i=1}^{d}(-1)^{k_i}a_{l_i}\binom{K+1}{k_i}\left(\int_{[-\pi,\pi]^d}\prod_{i=1}^{d}\cos\left(\frac{l_i}{k_i}\theta_i\right)g_f(\boldsymbol{\theta})\mathrm{d}\boldsymbol{\theta}\right).$$

It is not difficult to see that $|m_{l_1,k_1,\ldots,l_d,k_d}|$ can be upper bounded by a constant depending only on $d, K$ and $B$, but independent of $t$. This is because that the previous lemma shows $|a_{l_i}| \leq \frac{1}{\pi}$ and $g_f$ is upper bounded by a constant.

Now consider $c_{n_1,\ldots,n_d}$. Recall that

$$c_{n_1,\ldots,n_d} = (-1)^d \sum_{\substack{1 \le k_1,\ldots,k_d \le K+1 \\ 0 \le l_1,\ldots,l_d \le t \\ l_i = k_i \cdot n_i \forall i \in [d]}} m_{l_1,k_1,\ldots,l_d,k_d}.$$

We need to show that all $|c_{n_1,\ldots,n_d}|$ are upper bounded by a constant independent of $t$. Note that although each $l_i$ takes $t+1$ values, $l_i$ and $k_i$ must satisfy the constraint $l_i/k_i = n_i$. Since $k_i$ can take at most $K+1$ values, the number of $m_{l_1,k_1,\ldots,l_d,k_d}$ appeared in the summation is at most $(K+1)^d$. Therefore all $c_{n_1,\ldots,n_d}$ are bounded by a constant depending only on $d$, $K$ and $B$, and is independent of $t$.

$\square$

# C    The sparse grids algorithm

In this section we briefly describe the sparse grids numerical integration algorithm due to Gerstner and Griebel. (Please refer to [2] for a complete introduction.) We also specify a subroutine used by this algorithm, which is important for proving the running time.

Numerical integration algorithms dicretize the space and use weighted sum to approximate the integration. Traditional methods for the multidimensional case usually discretize each dimension to the same precision level. In contrast, the sparse grids methods, first proposed by Smolyaks [3], discretize each dimension to carefully chosen and possibly different precision levels, and finally combine many such discretization results. When the integrand has bounded mixed derivatives, as in our case that the integrand is in $C_B^K$, one can use very few grids in most dimension and still achieve high accuracy.

The sparse grids method is based on one dimensional quadrature (i.e., numerical integration). There are many candidates for one dimensional quadrature. In order to prove an upper bound for the running time, we choose the Clenshaw-Curtis rule [1] as the subroutine. This also makes the analysis simpler.

Let $h : [-1,1]^d \to \mathbb{R}$ be the integrand. Let $SG(h)$ be the output of the sparse grids algorithm. Let $l$ be the *level* parameter of the algorithm.

Let $\mathbf{k} = (k_1,\ldots,k_d)$ and $\mathbf{j} = (j_1,\ldots,j_d)$ be $d$-tuples of positive integers. Then $SG(h)$ is given as a combination of weighted sum:

$$SG(h) := \sum_{|\mathbf{k}| \le l+d-1} \sum_{j_1=1}^{m(k_1)} \cdots \sum_{j_d=1}^{m(k_d)} u_{\mathbf{k},\mathbf{j}} f(\mathbf{x}_{\mathbf{k},\mathbf{j}}). \qquad (2)$$

Below we describe $m(k_i)$, $\mathbf{x}_{\mathbf{k},\mathbf{j}}$ and $u_{\mathbf{k},\mathbf{j}}$ respectively.

1) For any $k \in \mathbb{N}$, $m(k) := 2^k$.

2) For each $\mathbf{k} = (k_1,\ldots,k_d)$ and $\mathbf{j} = (j_1,\ldots,j_d)$, define $\mathbf{x}_{\mathbf{k},\mathbf{j}} := (x_{k_1,j_1},\ldots,x_{k_d,j_d})$, and $x_{k_i,j_i}$ is the $j_i$th zero of the Chebyshev polynomial with degree $m(k_i)$. Denote by $T_{m(k_i)}$ the Chebyshev polynomial. Its zeros are given by the following formula.

$$x_{k_i,j_i} = \cos\left(\frac{(2j_i-1)\pi}{2m(k_i)}\right), \qquad j_i = 1,2,\ldots,m(k_i). \qquad (3)$$

3) Now we define the weights $u_{\mathbf{k},\mathbf{j}}$. First let $w_{k,j}$ be the weight of $x_{k,j}$ in the one-dimensional Clenshaw-Curtis quadrature rule given by

$$w_{k,1} = \frac{1}{(m(k)+1)(m(k)-1)},$$

$$w_{k,j} = \frac{2}{m(k)}\left(1+2\sum_{r=1}^{m(k)/2}{}' \frac{1}{1-4r^2}\cos\left(\frac{2\pi(j-1)r}{m(k)}\right)\right), \qquad \text{for } 2 \le j \le m(k), \quad (4)$$

where $\sum'$ means that the last term of the summation is halved.

Next, for any fixed $k$ and $j$, define

$$v_{(k+q),j} = \begin{cases} w_{k,j} & \text{if } q = 1 \text{ ,} \\ w_{(k+q-1),r} - w_{(k+q-2),s} & \text{if } q > 1, \text{ and for } r, s \text{ satisfying } x_{k,j} = x_{(k+q-1),r} = x_{(k+q-2),s} \text{ ,} \end{cases}$$

where $x_{k,j}$ is the zero of Chebyshev polynomial defined above.

Finally, the weight $u_{\mathbf{k},\mathbf{j}}$ is given by

$$u_{\mathbf{k},\mathbf{j}} = \sum_{|\mathbf{k}+\mathbf{q}| \leq l+2d-1} v_{(k_1+q_1),j_1} \cdots v_{(k_d+q_d),j_d},$$

where $\mathbf{k} = (k_1, \ldots, k_d)$ and $\mathbf{q} = (q_1, \cdots, q_d)$. This completes the description of the sparse grids algorithm.

## D    Proof of Lemma 4.4

We first give the result that characterizes the running time of the Gerstner-Griebel sparse grids algorithm in order to achieve a given accuracy.

**Lemma D.1.** *Let $h \in C_B^K([-\pi, \pi]^d)$ for some constants $K$ and $B$. Let $SG(h)$ be the numerical integration of $h$ using the sparse grids algorithm described in the previous section. Given any desired accuracy parameter $\tau > 0$, the algorithm achieves $\left| \int_{[\pi,\pi]^d} h(\theta)\mathrm{d}\theta - \mathrm{SG}(h) \right| \leq \tau$, with running time at most $O\left( \left(\frac{1}{\tau}\right)^{\frac{2}{K}} \left(\log \frac{1}{\tau}\right)^{3d+\frac{2d}{K}+1} \right)$.*

***Proof of Lemma D.1.*** Let $L = 2^{l+1} - 2$, where $l$ is the level parameter of the sparse grids algorithm. $l$ and $L$ will be determined by the desired accuracy $\tau$ later. In fact, $L$ is the maximum number of grid points of one dimension. By (2) it is easy to see that the total number of grid points, denoted by $N_l^d$, is given by

$$\begin{aligned} N_l^d &= \sum_{|\mathbf{k}| \leq l+d-1} m(k_1) \cdots m(k_d) \\ &= O(l^{d-1}L) \\ &= O(L(\log_2 L)^{d-1}). \end{aligned} \tag{5}$$

In [2] it is shown that the approximation error $\tau$ can be bounded by the maximal number of grid points per dimension as follows.

$$\tau = O(L^{-K}(\log L)^{(K+1)(d-1)}). \tag{6}$$

Next, let us consider the computational cost per grid point. Since we assume that $h(\mathbf{x})$ can be computed in unit time, and the zeros of Chebyshev polynomials can be computed according to (3), then computing the weights $u_{\mathbf{k},\mathbf{j}}$ dominates the running time. Fix $k \in \mathbb{N}$, consider $w_{k,j}$, $1 \leq j \leq m(k)$. From (4), it is not difficult to see that the set of $w_{k,j}$ can be computed by Fast Fourier Transform (FFT). Therefore the computation cost is $O(m(k) \log m(k))$. Some calculations yield that for a fixed $\mathbf{k}, \mathbf{j}$, the computational cost for $u_{\mathbf{k},\mathbf{j}}$ is $O(dL \log L)$. Combining this with (5) and (6) the lemma follows.

$\square$

Next we turn to prove Lemma 4.4. First, we need the following famous result.

**Lemma D.2.** *Let $m$ be a positive integer, let $\sigma(m)$ denotes the number of divisors of $m$, then for large $t$*

$$\sum_{m=1}^{t} \sigma(m) = t \ln t + (2c - 1)t + O(t^{1/2}),$$

*where $c$ is Euler's constant.*

To analyze the running time, we also need a result about the normalizing constant of the generalized Jackson kernel [4].

**Lemma D.3** ([4]). *Let*

$$J_{t,r} = \frac{1}{\lambda_{t,r}} \left( \frac{\sin(ts/2)}{\sin(s/2)} \right)^{2r},$$

*be the generalized Jackson kernel as given in Definition 4.1, and the normalizing constant $\lambda_{t,r}$ is determined by*

$$\int_{-\pi}^{\pi} J_{t,r}(s)\mathrm{d}s = 1.$$

*Then the following identity of the normalizing constant $\lambda_{t,r}$ holds*

$$\lambda_{t,r} = 2\pi \sum_{k=0}^{[r-r/t]} (-1)^k \binom{2r}{k} \binom{r(t+1)-tk-1}{r(t-1)-tk}. \tag{7}$$

Now we are ready to prove Lemma 4.4.

***Proof of Lemma 4.4.*** Assume that the error induced by the sparse grids algorithm is at most $\tau$ per integration. That is, for every $\mathbf{k} = (k_1, \ldots, k_d), \mathbf{l} = (l_1, \ldots, l_d)$

$$\left| \int_{[-\pi,\pi]^d} \prod_{i=1}^{d} \cos\left( \frac{l_i}{k_i}\theta_i \right) g(\boldsymbol{\theta})\mathrm{d}\boldsymbol{\theta} \right| \leq \tau.$$

Then

$$\sup_{n_1,\ldots,n_d} |c_{n_1,\ldots,n_d} - \hat{c}_{n_1,\ldots,n_d}| \leq \sup_{n_1,\ldots,n_d} \left| \sum_{l_i/k_i=n_i} \prod_{i=1}^{d} (-1)^{k_i} \binom{K+1}{k_i} a_{l_i} \right| \cdot \tau.$$

By Lemma B.1, $|a_{l_i}| \leq \frac{1}{\pi}$. We obtain that

$$\sup_{n_1,\ldots,n_d} |c_{n_1,\ldots,n_d} - \hat{c}_{n_1,\ldots,n_d}| \leq M \cdot \tau, \tag{8}$$

for some constant $M$ independent of $t$.

Similarly, we have

$$\left\| I_{t,K}^d(g) - \hat{I}_{t,K}^d(g) \right\|_{\infty} \leq O(t^d\tau). \tag{9}$$

Since in the statement of the lemma the desired approximation error is $O(t^{-K})$, we have

$$\tau = t^{-(K+d)}. \tag{10}$$

It is also clear that

$$\max_{n_1,\ldots,n_d} |\hat{c}_{n_1,\ldots,n_d} - c_{n_1,\ldots,n_d}| = o(1), \quad \text{as } t \to \infty.$$

Now let us consider the computation cost. Recall that the kernel $H_{t,r}$ is an even trigonometric of degree at most $t$:

$$H_{t,r}(s) = a_0 + \sum_{l=1}^{t} a_l \cos ls, \tag{11}$$

where $H_{t,r}(s) = J_{t',r}(s)$ and $J_{t',r}$ is the generalized Jackson kernel given in Definition 4.1. First we need to compute the value of the linear coefficient $a_l$ of $H_{t,r}$. By Lemma D.3, one can compute the linear coefficients $a_l$ by solving a system of $t+1$ linear equations. That is, we choose an arbitrary $t+1$ points in $[-\pi, \pi]$ and solve (11), since we can compute the value of $H_{t,r}(s)$ directly based on the value of $\lambda_{t,r}$. Clearly, the running time is $O(t^3)$.

Having $a_{l_i}$, let us consider the computational cost for calculating $\hat{c}_{n_1,\ldots,n_d}$. According to Lemma D.1, the running time for the sparse grids algorithm to compute one integration is $O\left(\left(\frac{1}{\tau}\right)^{\frac{2}{K}}\left(\log(1/\tau)\right)^{3d+\frac{2d}{K}+1}\right) = O\left(t^{\frac{2(K+d)}{K}}\text{polylog}(t)\right)$.

Since we only need to compute the integration when $l_i | k_i$ for all $i \in [d]$, by Lemma D.2 the number of integrations to compute is at most

$$(K + 1 + \sigma(1) + \ldots + \sigma(t))^d = O\left((t\log t)^d\right).$$

Thus the total time cost for all numerical integration is $O\left(t^{(1+\frac{2}{K})d+2}\text{polylog}(t)\right)$. Since

$$(1 + \frac{2}{K})d + 2 \geq 3,$$

the computation time for obtaining the coefficients $a_l$ in $H_{t,r}$ is dominated by the running time of the sparse grids algorithm. It is also easy to see that all other computation costs are dominated by that of the numerical integration. The lemma follows.

$\square$