[Reviews · NeurIPS 2013]

Submitted by Assigned_Reviewer_6

Efficient Algorithm for Privately Releasing Smooth Queries
-------------------------------------------------------------------------------
The authors address the problem of answering a large set of queries in a differentially private manner. The set in question in this paper is the set of function with bounded partial derivatives up to order K. The authors' technique mimics the work of Thaler et al (ICALP 2012) only the authors decompose the queries not into regular polynomials (Chebyshev polynomials in the case of Thaler et al), but rather to trigonometric polynomial in this case. The bulk of the work is indeed to show that the abovementioned set of queries can be well-approximated by trigonometric polynomials. Having established that, adding Laplace noise to each monomial suffices to guarantee differential privacy.

The paper has all the ingredients of a great paper -- the authors tackle a classical problem using an original approach, prove something far from trivial, and Table 1 shows what one would expect to see: as K increases, the set of queries is more restricted, and so the error bounds for the queries get tighter. Furthermore, I find it to be very well written, even the more complicated mathematical parts.

After the authors' rebuttal I now see that there are situations, though admittedly-- somewhat limited, in which this mechanism out-performs the naive straw-man algorithms of MW [HR10, GRU12] and Exponential mechanism [MT07, BLR08] in terms of error and/or running time. I therefore support acceptance.

I suggest the authors do the following:

1. Mention the straight-forward techniques at the intro, and show under certain situations (i.e., smoothness bound > 2*dimension) they get error of o(n^{1/2}), in time o(n^{d/2}).

2. It will also be helpful to compare to the bounds you get using the inefficient BLR-mechanism (in essence, the authors' technique in the given paper bounds the ``effective'' number of queries in C_B^K).

3. If the authors can incorporate the results regarding S non-zeros derivatives into the paper -- all the better.
Summary: The authors address the problem of answering a large set of queries in a differentially private manner, focusing on queries with bounded partial derivatives and using trigonometric polynomial. Their technique applies in a limited setting, but there it outperforms other existing techniques.

Submitted by Assigned_Reviewer_7

NOTE: Due to the number of papers to review and the short reviewing time, to maintain consistency I did not seek to verify the correctness of the results. As a result, I generally ignored any proofs or additional details in the supplementary material. This review should be considered as a review of the main paper alone.

SUMMARY: The authors propose a new method for approximating a database in a differentially private manner when the database will be used to answer queries with certain smoothness properties. The method appears to be correct but is unclear if it is primarily theoretical interest or if it is practically relevant.

STRENGTHS:
- This is a new approach to database approximation for real data, a topic which has not received as much attention in the differential privacy literature.
- The application of polynomial approximation is novel with respect to this problem (although similar ideas for related problems have been proposed before).

WEAKNESSES:
- It is unclear if the proposed method actually "works" in any meaningful sense. The authors are not very clear on this point. Since there is no experimental verification, it should be more clear that this is a primarily theoretical investigation.
- It is not clear from the statement of the results if bounds are surprising, to be expected, or if there is a hope of matching lower bounds. This makes it hard to evaluate the "impact" of the result.


COMMENTS:
- Does it make sense that the smoothness of the function should depend on the data dimension? In particular, the fact that $K/d$ is the relevant quantity is not very intuitive.
- The authors seem to want to make a big deal out of the fact that their approach allows "infinite number of queries," but this is pretty obvious because they are not really looking at a query model but instead a differentially private approximation of the data. In the discrete setting, if one makes a differentially private histogram, one gets the same thing.
- I was hoping for more of a comparison with [18], which (as I recall) also looks at real-valued data and derives estimators based on a quantization approach. The authors cite this paper but do not provide any context.
- Indeed, the whole "name of the game" here is density estimation, which is a well-studied area in statistics and machine learning. What does this literature have to say about the approach suggested by the authors?
- Experiments, evaluation? This seems like a nice recipe for approximation but it is a little hard to see if, e.g. it will work at all for d of "practical" interest. Indeed some examples of useful smooth queries could be nice. Is this an interesting sub-class of problems that could be used as an example?

UPDATE AFTER REBUTTAL:
* After the response and discussion I remain sanguine about this paper. However, I would like many of the details from the author's rebuttal to appear in the manuscript -- this will help clarify the presentation and might actually facilitate some discussion at NIPS.
* The authors did not really address many of my comments, which is somewhat disappointing. I think an example of interesting smooth queries (e.g. in the introduction) will help ground the theory. Otherwise many of the more applied folks interested in privacy will say "oh, the authors show an algorithm for some version of smoothness for which they give no interesting examples except references to [23] and [28]." A concrete example of an interesting smooth query that is relevant to the NIPS audience will make this paper more accessible.
Summary: The authors propose a new method for approximating a database in a differentially private manner when the database will be used to answer queries with certain smoothness properties. The method appears to be correct but is unclear if it is primarily theoretical interest or if it is practically relevant.

Submitted by Assigned_Reviewer_8

This paper presents an algorithm for releasing the answers to a set of smooth queries on a database, in a manner that preserves differential privacy. Smooth queries are those whose partial derivatives up to a particular order are all bounded.

The basic idea is essentially that smooth functions can be reconstructed from a limited number of basis functions, so outputting a differentially private such basis allows one to output differentially private smooth functions. The result doesn't have depth on the privacy side---the techniques and proofs are completely standard. And the "basis" result as I'm calling it (trigonometric polynomial approximation), is not new. So the contributions of the paper seem to be in the technical details of ensuring the approximations have small enough coefficients and in the idea of applying differential privacy in this setting. It's a nice idea, but the overall contribution doesn't excite me. Perhaps a reader with a deeper interest in (approximations of) smooth functions would appreciate it more.

A few other comments:
The definition of accuracy as written is confusing; it would help to clarify that you're taking probability over the randomness of the mechanism.

The paper needs to be proofread; there are lots(!) of minor grammatical errors.
Summary: Summary: There is a nice idea at the core of this paper, but I am not excited enough about the overall contribution.
Author Feedback

Author rebuttal: To reviewer 1 (Assigned Reviewer 6)

[Comment] The paper has a major fault. One can discretize [-1,1]^d and use existing algorithms which then outperform the proposed approach.

[Answer] We thank the reviewer for raising technical questions. However the reviewer seems to make a miscalculation (see below for detail). Please note that:

1) We already discussed these discretization based methods and compared them to our approach in the paper. Please see the 3rd paragraph in Section 5.

2) Our algorithm significantly outperforms discretization based methods both in accuracy and running time for queries with high order smoothness.

First let us compare the performance of our algorithm with the best discretization based method (e.g., discretization + MW). Consider the case that the queries have order of smoothness K which is large compared to d.

- For the discretization based approaches, the best possible accuracy is n^{-1/2}. To achieve such an accuracy, the running time of the algorithm is at least n^{d/2}. (See line 423-428 in the paper.)

- For our algorithm, the accuracy is nearly n^{-1}; and the running time is n^c for some c << 1. (See the 3rd row in Table 1.)

Thus, for highly smooth queries, our algorithm has an error much smaller than the standard n^{-1/2}, which is inherent to all previous mechanisms for answering general linear queries. Moreover, the running time of our algorithm to achieve the best accuracy is sublinear, much better than n^{d/2} for discretizaiton based methods.

To see why the reviewer's conclusion is different to ours, we point out a mistake in reviewer's comment. The comments are that one can discretize [-1,1]^d to a constant precision (denoted by gamma); and on the discretized data, existing algorithms are faster than our algorithm. However, as the reviewer also admitted, the error of such an approach is a constant, which means that the error does not decrease even if more data is collected. To reduce the error to o(1), the reviewer then sets the discretization precision gamma polynomially small.

But here comes the problem: if gamma gets small, the running time of the discretization based algorithm grows: it scales as (1/gamma)^d. To be concrete, if the discretization precision is gamma, the total number of grids is (1/gamma)^d, and the running time of the best existing dp algorithm at least equals to the number of grids. Also note that the error of the discretization based method is gamma + n^{-1/2}, which is the sum of the dicretization error and the error induced by the dp algorithm. So the best possible accuracy is n^{-1/2} (by setting gamma = n^{-1/2}), and the corresponding running time is n^{d/2}, both are outperformed by our algorithm for highly smooth queries.

It is also worth pointing out that both the accuracy and running time of the discretization based methods are independent of the order of smoothness K as long as K >= 1; because the algorithms only use the first order smoothness. In contrast, our algorithm fully exploits the smoothness and has better performance when K is large.

In sum, for highly smooth queries, the discretization based algorithms run in time n^{d/2} to achieve an error of n^{-1/2}; while our algorithm runs in sublinear time and achieves an error nearly n^{-1}.



[Comment] Suppose the derivatives are sparse. Only S of the d^K partial derivatives are non-zero. Can this allow a non-constant d?

[Answer] We appreciate your suggestion very much. This is an interesting problem. We conducted some preliminary analysis. Here are a few results.

1) The simplest case is that the sparsity parameter S is a constant. Then d can be as large as n^{Omega(1)}. The performance of the algorithm only has a minor decrease. For very smooth queries, the accuracy is still significantly better than n^{-1/2}, and the running time is sublinear.

2) More interesting is the case that S is larger than any constant. We found that a more refined sparsity parameter S_K is crucial. Here S_K is the number of non-zero Kth order partial derivatives. We are able to show that if S_K is a constant, then S can be as large as (log n)^{1/2}, and d can be as large as 2^{(log n)^{1/2}}. The performance of (a slightly variant of) our algorithm does not change too much.

For more general cases, we currently do not know the answer. We think that this sparsity problem deserves a deep study.


[Comment] The setting considered is very limited: the dimension d is a constant.

[Answer] When studying differential privacy on Euclidean space R^d, it is common in literature to assume a constant dimension d. Please see for example the references [2, 5, 18, 29] and Dwork and Lei (STOC 2009). We follow this convention.



To Reviewer 2 (Assigned Reviewer 7)

Thank you for the comments. This paper is mainly a theoretical study. Experimental evaluation is surely our future work.

Our results state that for highly smooth queries our algorithm can achieve an accuracy of nearly n^{-1}, while previous approaches have n^{-1/2} at best. To answer a query, the running time of our algorithm is sublinear in n, much better than n^{d/2} for existing methods.




To Reviewer 3 (Assigned Reviewer 8)

[Comment] The result does not have depth on the privacy side.

[Answer] Privacy cannot be separated from accuracy and efficiency. There are many differentially private algorithms for which, like ours, proving privacy is easy but proving bounds on the error and running time is difficult. Please see also (Blocki, Blum, Datta, and Sheffet, FOCS 2012) for a discussion about this common phenomenon. We do not think this means a lack of depth on the privacy side.


[Comment] The definition of accuracy as written is confusing. It appears to evaluate the probability that there exists such a query, but that is not what you want.

[Answer] This is the standard definition of worst case accuracy. Please see, e.g., refs [2, 10, 16] in the paper.